# Study of Hybrid Composite Joints with Thin-Ply-Reinforced Adherends

**DOI:** 10.3390/ma16114002

**Published:** 2023-05-26

**Authors:** Farin Ramezani, Ricardo J. C. Carbas, Eduardo A. S. Marques, Lucas F. M. da Silva

**Affiliations:** 1Instituto de Ciência e Inovação Em Engenharia Mecânica e Engenharia Industrial (INEGI), Rua Dr. Roberto Frias, 4200-465 Porto, Portugal; 2Departamento de Engenharia Mecânica, Faculdade de Engenharia (FEUP), Universidade Do Porto, Rua Dr. Roberto Frias, 4200-465 Porto, Portugal

**Keywords:** composite joints, thin ply, single lap joints

## Abstract

It has been demonstrated that a possible solution to reducing delamination in a unidirectional composite laminate lies in the replacement of conventional carbon-fibre-reinforced polymer layers with optimized thin-ply layers, thus creating hybrid laminates. This leads to an increase in the transverse tensile strength of the hybrid composite laminate. This study investigates the performance of a hybrid composite laminate reinforced by thin plies used as adherends in bonded single lap joints. Two different composites with the commercial references Texipreg HS 160 T700 and NTPT-TP415 were used as the conventional composite and thin-ply material, respectively. Three configurations were considered in this study: two reference single lap joints with a conventional composite or thin ply used as the adherends and a hybrid single lap. The joints were quasi-statically loaded and recorded with a high-speed camera, allowing for the determination of damage initiation sites. Numerical models of the joints were also created, allowing for a better understanding of the underlying failure mechanisms and the identification of the damage initiation sites. The results show a significant increase in tensile strength for the hybrid joints compared to the conventional ones as a result of changes in the damage initiation sites and the level of delamination present in the joint.

## 1. Introduction

Composites usually consist of two main components known as the matrix, which provides the cohesion of the material, and the reinforcement, such as fibres, which provides the material with its strength and stiffness [1]. The use of carbon-fibre-reinforced polymer (CFRP) materials in multiple industrial applications is continuously increasing [2,3,4,5,6,7], leading to the use of a vast range of composite materials for the design and manufacture of high-performance composite products such as vehicle structures, sporting goods, etc. [2,8]. However, since the strength of the matrix is at least an order of magnitude lower than the strength of the reinforcement, the loads applied in a perpendicular direction to the reinforcement are almost exclusively carried by the low-strength matrix. This results in the onset of delamination, which can lead to the rapid degradation of the mechanical performance of the structure and cause premature failure [9,10,11,12]. Accordingly, multiple studies have investigated methods for adhesive layer modification [13,14] or composite laminate modification to delay delamination in a composite joint [15,16,17], such as the use of fibre metal laminates (FML), composite laminates with toughened layers [18,19], glass fabric reinforcement [20], the use of Z-pins [21,22], 3D weaving [23], stitching [23], braiding [24], or even the adoption of additional thermoplastic inter-plies [25]. However, the significant complexity of these techniques often restricts their usage. Furthermore, such methods normally require the implementation of at least one additional production step [26,27] and thus increase the costs associated to the production process.

Recent advancements in composite manufacturing techniques have led to the development of spread-tow technology [28], which results in flat and straight plies with a more homogeneous fibre distribution and smaller resin-rich regions [29,30]. In this case, a dry ply thickness as low as 0.02 mm can be achieved. Generally, plies with a thickness below 100 μm are known as thin plies [31]. By reducing the thickness of a single layer, the number of possible total layers and therefore the degrees of freedom in design are increased [32]. This also results in a larger number of interfaces in thin-ply laminates, lowering the shear stresses [32,33]. Moreover, thin-ply laminates are known for their ability to delay the onset of the matrix damage mechanisms and suppress transverse microcracking [31] and free edge delamination [32,34] for static, fatigue, and impact loadings. Due to their superior damage and delamination resistance, thin-ply laminates could exhibit higher interlaminar shear properties [35] and strain energy [36] compared to conventional plies. Therefore, thinner composite plies are acknowledged to have higher in situ transverse strength [33]. Nonetheless, the properties of the laminate can still rapidly deteriorate after the onset of damage, leading to premature structural failure [11]. Thin plies are now seen as a promising approach to improve the performance of adhesively bonded CFRP, mainly due to their ability to enhance the off-axis performance of composites and postpone delamination [36]. Moreover, studies have shown that through the use of thin plies in a structural joint, the damage location in the composite moves from the adhesive interface towards the mid-thickness of the composite adherends [36], mainly due to the in situ effect [37]. Through-thickness reinforcement can effectively provide improved interlaminar strength and delamination resistance while producing a more integrated composite structure [38].

A previous study by the authors [39] showed that replacing layers of a conventional composite in a unidirectional laminate with layers of thin ply can increase the composite’s strength under transverse tensile loading. The authors postulate that this is due to the increase in laminate ductility, which could postpone the delamination under transverse tensile loads. Moreover, experimental observation clearly demonstrated that the presence of thin plies acts as a barrier against crack propagation. It was shown that the use of 25% (corresponding to the optimum amount) thin ply per total thickness of the laminate (12.5% on each top) increased the transverse tensile strength considerably. Figure 1a shows the studied configuration for a reference conventional composite: thin ply and the optimum hybrid laminate. Figure 1b illustrates the experimentally obtained failure loads for the mentioned configurations.

The current study seeks to further investigate this topic, quantifying the performance of a hybrid composite single lap joint and analysing the effect of reinforcing the unidirectional conventional composite adherend using thin ply. In this work, “HS 160 T700” and “NTPT-TP415” are used as a conventional composite and thin-ply material, respectively. The tests were recorded using a high-speed camera to determine the location of damage initiation. Afterwards, the failure surface of specimens was analysed and measured via image analysis, allowing for an accurate estimation of the delaminated area. It was found that the use of hybrid single lap joints reinforced with thin-ply layers results in a considerable increase in the joint strength compared to the reference conventional composite joint. Numerical models were also created via cohesive zone modelling, allowing for the accurately replication and description of the experimentally determined failure processes.

## 2. Experimental Details

### 2.1. Adhesive

The adhesive used in this work was an epoxy structural adhesive, supplied in film form, with the commercial reference Scotch Weld AF 163-2k (3M, Saint Paul, MN, USA) [40]. The adhesive was cured following the manufacturer’s recommendations at 130 °C for 2 h. The mechanical properties of the AF 163-2k adhesive are presented in Table 1.

### 2.2. Adherend

#### 2.2.1. Conventional Composite

The materials used in the studied configurations were chosen to be representative of a possible application within the aerospace sector. Accordingly, a unidirectional prepreg carbon–epoxy composite with a ply thickness of 0.15mm was selected, with the commercial reference Texipreg HS 160 T700 (Seal Spa, Legnano, Italy). This is an orthotropic material whose mechanical properties are presented in Table 2. The elastic mechanical properties of the conventional composite correspond to the orientation of a 0° composite ply (1 and 2 are defined as the fibre and transverse directions). Moreover, the cohesive properties of the conventional composite’s resin are presented separately in Table 3.

#### 2.2.2. Thin-Ply

A unidirectional, 0° oriented carbon–epoxy prepreg composite with a ply thickness of 0.075 mm was selected for use in this work, serving as the thin-ply material. This material has the commercial reference NTPT-TP415 (North thin ply technology, Zory, Poland). The elastic orthotropic and cohesive properties for the thin ply, characterised by the authors in a previous work [44], are presented in Table 4 and Table 5, respectively.

### 2.3. Single Lap Joint Manufacturing

Single lap joints (SLJs) were manufactured with the geometry shown in Figure 2. The width for all specimens under consideration was set at 15 mm.

The manufacturing process for the reference conventional composite and thin-ply adherends began with a layer-by-layer stacking of the conventional composite and thin-ply prepregs respectively, until the desired adherend thickness was attained (3.6 mm). In this case, 24 and 48 layers of conventional composite and thin-ply prepreg were used, respectively. For the hybrid (25% thin ply) adherends, 6 plies of conventional composite were replaced by 12 plies of thin ply on the adherend tops (6 layers of thin ply on each adherend top). The joints were then bonded by applying an additional layer of adhesive between the adherends. A mould was used to ensure the thickness of the adherends and adhesive. A mould-release agent was used to ensure easy debonding of the specimen from the mould after curing. It was observed that the curing sequence, i.e., curing the adhesive and substrate composite plies in one cure (co-curing) or in two separate cures, had no significant effect on the mechanical properties of the joint for the AF163-2k adhesive. Therefore, a one-step curing manufacturing method was preferred to simultaneous reducing manufacturing time and energy usage. Accordingly, the joint was co-cured at 130 °C for two hours, following the manufacturer’s recommended procedure. A schematic design of the reference conventional composite, thin-ply, and hybrid (25% thin-ply) single lap joints are shown in Figure 3.

### 2.4. Testing Condition

The SLJs were tested using an Instron 8801 servo hydraulic testing machine with a load cell of 100 kN and at a constant crosshead speed of 1 mm/min. All tests were performed under laboratory ambient conditions (room temperature of 24 °C, relative humidity of 55%). Four repetitions were performed for each configuration under analysis.

## 3. Experimental Results

### 3.1. Load–Displacement Curve

Figure 4 shows representative, experimentally obtained load–displacement curves for the studied configurations. The hybrid (25% thin ply) joint presented the highest failure load, with an increase in joint strength of approximately 90% compared to the reference conventional composite configuration.

### 3.2. Damage Initiation

A high-speed camera was used to record the specimens under load, seeking to determine whether the damage initiation occurred first in the adhesive layer or within the adherend. A Chronos 1.4 high speed camera was used, recording at 5000 frames per second. Figure 5 presents the images at damage initiation for each configuration. The adhesive and adherend boundaries were roughly defined by correlating the known specimen’s dimensions and the equivalent image pixels. As can be seen in Figure 5, in the reference conventional composite and the thin-ply, damage initiation occurred in the composite adherend. In contrast, for the hybrid (25% thin ply) joint, the damage initiation occurred in the adhesive layer.

### 3.3. Delamination

Digital images of the failure surface were analysed in order to obtain the delamination ratio for each configuration. The open-source software IC Measure was used to calculate the delamination area for each joint. The delamination ratio is defined as the delamination area divided by the total bonded area, as presented in Equation (1). It should be noted that the total bonded area was constant and equal to 375 ${\mathrm{mm}}^{2}$ for all configurations. Figure 6 provides representative images of the failure surface for all configurations. The representative delamination area from Figure 6 and the average delamination area are presented in Table 6. The reference conventional composite joint shown in Figure 6 presents delamination of about 51%, while the hybrid (25% thin ply) configuration presents delamination of about 29%. In contrast, around 80% of the total area was observed to have suffered delamination in the reference thin-ply joint.
(1)$$Delamination\;ratio\;(\%)=\frac{Delamination\;area}{Total\;bonded\;area}$$

### 3.4. Microscopic Images

The failure surfaces of the reference conventional composite and hybrid (25% thin ply) joint were analysed using a ZEISS AXIOPHOT microscope. According to the microscopic images presented in Figure 7, multiple fibre breakages [45] and fibre pull-outs [46] were observed on the failure surface of the reference conventional composite failure surface. These failure mechanisms were not observed in the failure surface of the hybrid (25% thin ply) joint under analysis.

## 4. Numerical Study

### 4.1. Load–Disploacement Curve

A two-dimensional statically loaded model was used to simplify the problem under analysis and reduce the computational time. Boundary conditions were defined as shown in Figure 8. The left end of the joint was fixed while a displacement was applied in the right end to replicate the testing fixtures. A cohesive zone model (CZM) was used to model the adhesive behaviour, employing four node elements: cohesive quadrilateral elements. Non-linear geometrical effects were included. Solid cohesive elements following triangular traction separation laws were applied to the adhesive layers of the model to simulate damage evolution (damage initiation and propagation). Cohesive behaviour was specified directly in terms of a traction–separation law, which has been shown to be suitable to represent delamination in composite laminates [47]. Therefore, a similar CZM was introduced into the composite material (conventional composite or thin-ply) to model delamination due to the experimental failure mode obtained. These interlaminar cohesive element layers were placed in between elastic homogeneous sections (see Figure 9) and effectively simulated the possible debonding between the plies of composite. The CZM layers were placed at a distance of 0.15, 0.37, and 0.13 mm from the interface of the adherend and adhesive layer for the conventional composite, thin-ply, and hybrid (25% thin ply) joints, respectively. This distance roughly corresponded to the experimentally measured distance of the delamination plane from the adhesive layer. The thickness of the cohesive layer matched the thickness of one equivalent composite ply (0.075 mm for thin ply and 0.15 mm for the conventional composite).

Double and single biased mesh distributions were considered in the *x* direction (see Figure 10) for the bondline and the adherends, respectively. The minimum and maximum sizes for the mesh were considered 0.2 and 0.5 mm respectively. However, a uniform mesh distribution with the size of 0.5 mm was considered for the end tabs (in the *x* direction). Moreover, a uniform distribution through the mesh thickness (*y* direction) was considered for all models with a mesh size of 0.2 mm. Figure 10 illustrates the mesh distribution mentioned above. As a result, around 15,000 elements were generated for each numerical model. Figure 11 presents the numerical load–displacement curves obtained for all configurations. As shown, the numerical results are in good agreement with the experimentally obtained load–displacement curves.

### 4.2. Damage

The same model was used to determine the damage initiation and its propagation mode for each configuration under analysis. According to the numerical results presented in Figure 12, the damage for the reference conventional composite and thin-ply joint initiated in the composite (conventional composite and thin-ply joints, respectively). For the hybrid (25% thin ply) joint, damage initiation occurred in the adhesive layer. The loads at damage initiation for the mentioned configurations were 3.6, 6.9, and 7.3 kN, respectively. It should be noted that the damage generated in the configurations at the equivalent numerically obtained failure loads illustrates that delamination is expected to be the final failure mode for all configurations under analysis (see Figure 13).

## 5. Discussion

The use of hybrid, adhesively bonded composite joints reinforced with thin plies increases the tensile strength compared to that of reference joints manufactured using only a conventional composite. Figure 14 presents the average failure load obtained for the reference conventional composites, thin-ply, and hybrid (25% thin ply) single lap joints. An increase of about 90% in the failure load was obtained for the hybrid (25% thin ply) joints. Although similar failure loads were obtained for the hybrid (25% thin ply) and reference thin-ply joint, it must be mentioned that the manufacturing process for a hybrid (25% thin ply) joint costs less and is less time-consuming compared to the process of manufacturing the reference thin-ply joint. Moreover, experimental observation illustrates that the delamination ratio decreases considerably while using hybrid composite joints reinforced with thin- ply compared to the reference conventional composite and thin-ply single lap joint. The average delamination ratio obtained for this configuration can be found in Figure 15. According to the numerical and experimental study, the damage initiation location depends on the joint configuration. Damage first occurs in the adherend in the reference conventional composite and thin ply, but it initiates in the adhesive layer for the hybrid (25% thin ply) joint. The initiated damage propagates as a combination of delamination and cohesive failure for all configurations, but a lower level of delamination was obtained for the hybrid joint. A schematic representation of the mentioned failure mechanism could be found in Figure 16.

## 6. Conclusions

This study investigated the mechanical performance of composite single lap joints using toughened adherends reinforced with thin plies. A numerical and experimental study was performed accordingly. The main conclusions drawn from this work are as follows:An increase of approximately 90% in the failure load was found for the hybrid joint reinforced with thin ply when compared to the reference conventional composite joint.According to the experimental observation, damage initiation occurs in the adherend for the reference conventional composite and thin-ply joint, while for the hybrid (25% thin ply) joint, damage initiation occurs in the adhesive layer.Damage propagates as a combination of delamination and cohesive failure for all configurations. However, a more limited amount of delamination was obtained for the hybrid joint.Microscopic images of the bond line allowed for the identification of multiple fibre breakages and fibre pull-outs on the failure surface of the reference conventional composite configuration. In contrast, the fibres were still intact and well-aligned in the failure surface of the hybrid joint.The configurations under analysis were modelled numerically, and a good agreement was obtained between the numerical and experimental results, allowing for a precise representation of the damage initiation and failure processes.

## Figures and Tables

**Figure 1 materials-16-04002-f001:**
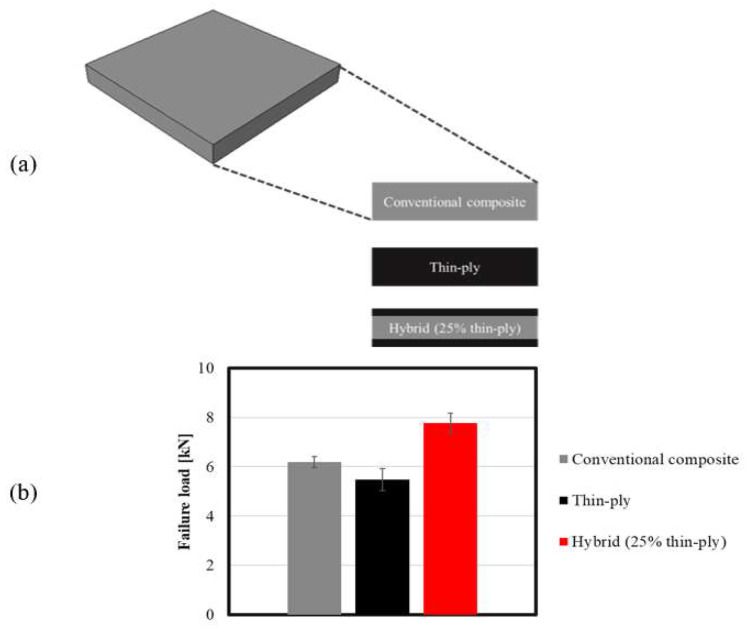
(**a**) Schematic design for conventional composite, thin-ply, and hybrid (25% thin ply) laminates and (**b**) summary of the experimentally obtained failure load for unidirectional reference conventional composite, thin-ply, and hybrid (25% thin ply) laminates. Adopted from [39].

**Figure 2 materials-16-04002-f002:**
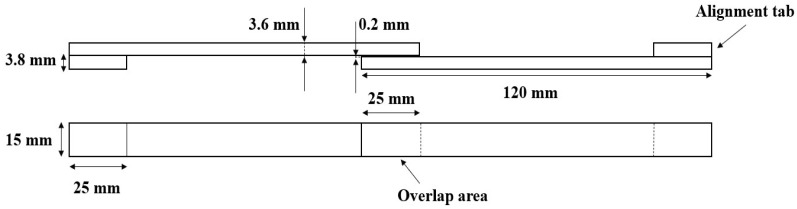
Single lap joint geometry.

**Figure 3 materials-16-04002-f003:**
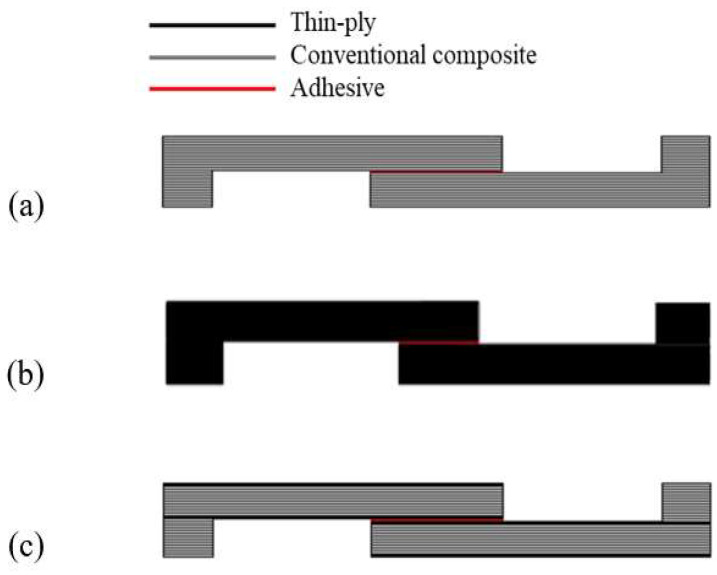
Schematic design of (**a**) conventional composite, (**b**) thin-ply, and (**c**) hybrid (25% thin-ply) joints.

**Figure 4 materials-16-04002-f004:**
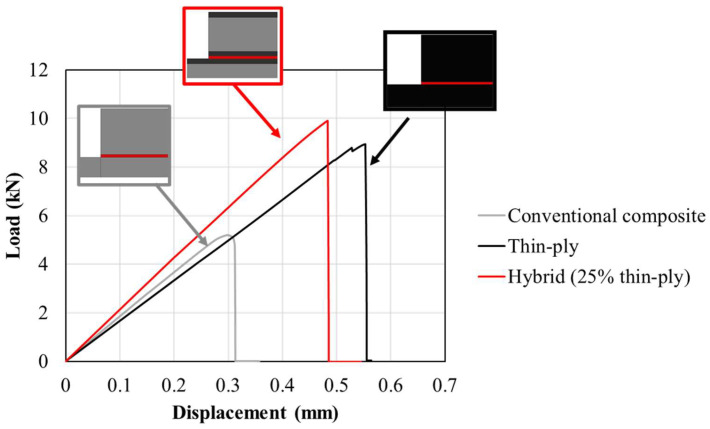
Representative load–displacement curves for reference conventional composite, thin-ply, and hybrid (25% thin ply) joints.

**Figure 5 materials-16-04002-f005:**
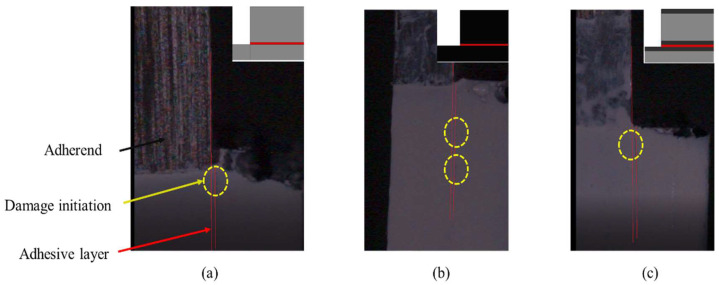
Damage initiation for (**a**) conventional composite, (**b**) thin-ply, and (**c**) hybrid (25% thin ply) joints.

**Figure 6 materials-16-04002-f006:**
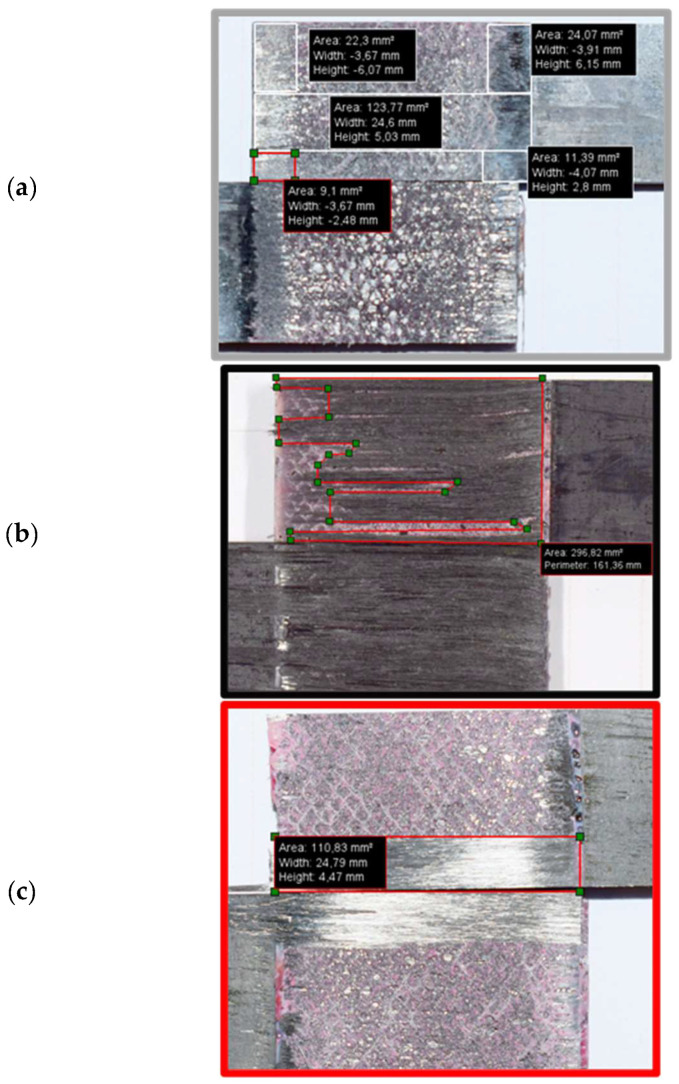
Representative images of failure surface of (**a**) reference conventional composite, (**b**) thin-ply, and (**c**) hybrid (25% thin ply) joints.

**Figure 7 materials-16-04002-f007:**
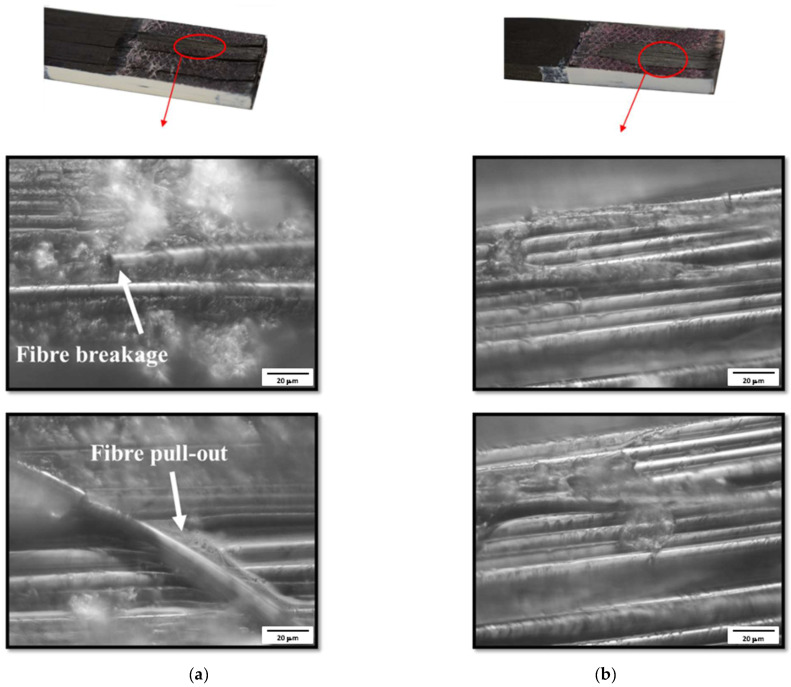
Microscopic images of (**a**) conventional composite and (**b**) hybrid (25% thin ply) joints.

**Figure 8 materials-16-04002-f008:**
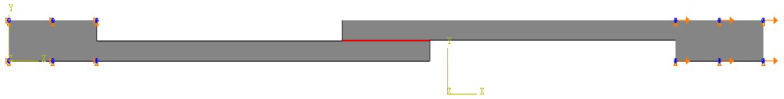
Boundary condition of simulated single lap joint.

**Figure 9 materials-16-04002-f009:**
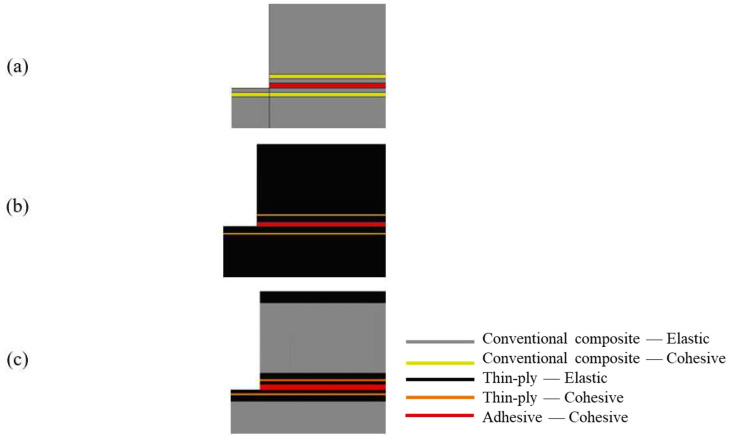
Assigned mechanical properties for (**a**) conventional composite, (**b**) thin-ply, and (**c**) hybrid (25% thin ply) joints.

**Figure 10 materials-16-04002-f010:**
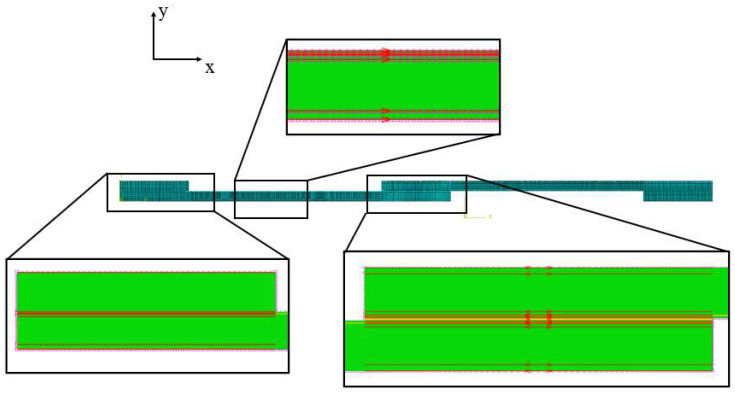
Mesh distribution for numerical models.

**Figure 11 materials-16-04002-f011:**
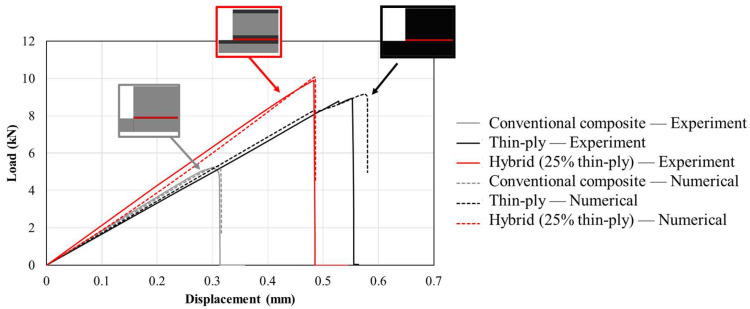
Comparison of numerically obtained load–displacement curves for conventional composite, thin-ply, and hybrid (25% thin ply) joints with the representative experimental results.

**Figure 12 materials-16-04002-f012:**
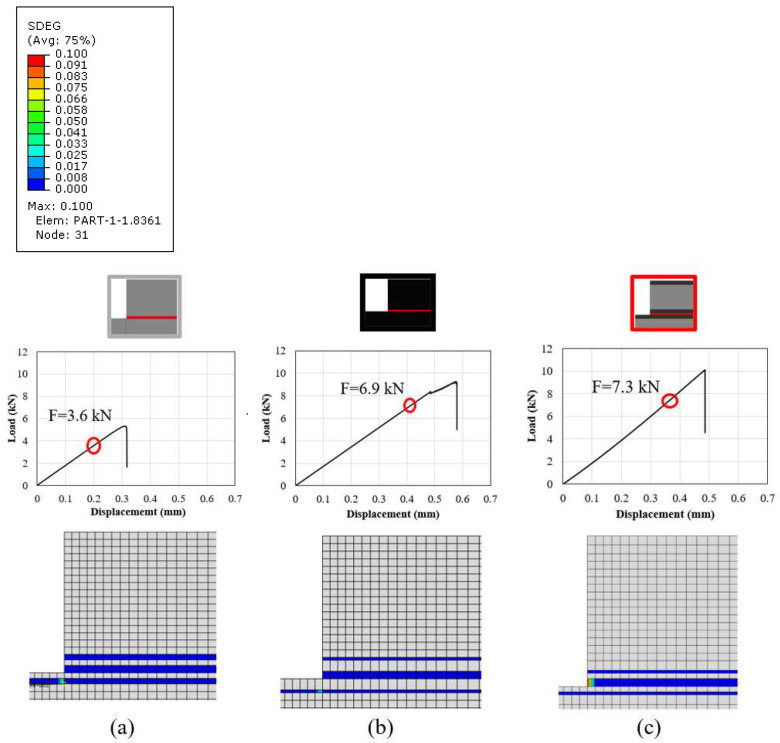
SDEG at damage initiation for (**a**) conventional composite, (**b**) thin-ply and (**c**) hybrid (25% thin ply) joints (equivalent loads for each configuration are 3.6, 6.9, and 7.3 kN, respectively).

**Figure 13 materials-16-04002-f013:**
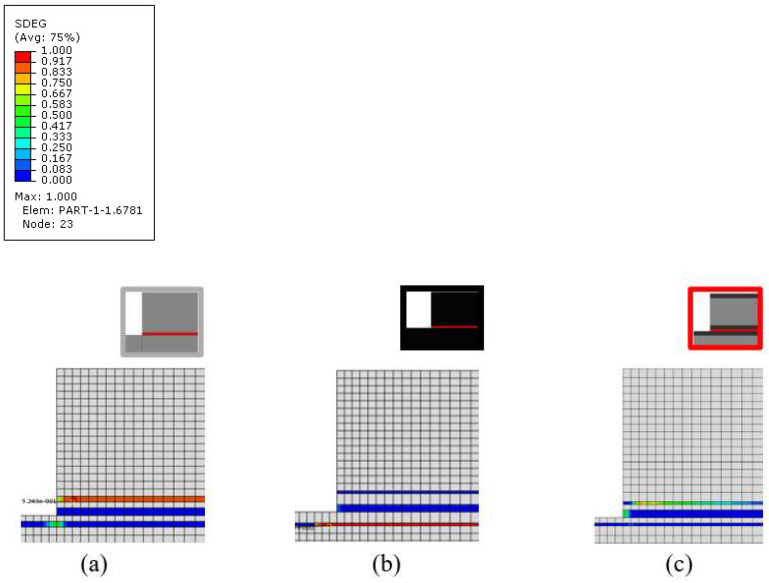
SDEG at failure load for (**a**) conventional composite, (**b**) thin-ply and (**c**) hybrid (25% thin ply) joints (equivalent loads for each configuration are 5.4, 9.2, and 10.0 kN, respectively).

**Figure 14 materials-16-04002-f014:**
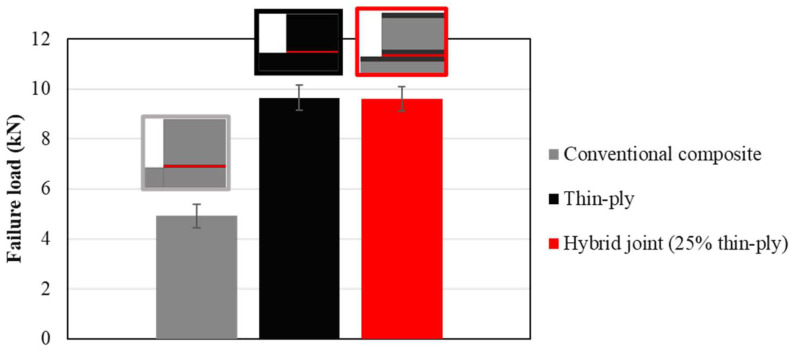
Average failure loads obtained experimentally for reference conventional composite, thin-ply, and hybrid (25% thin ply) joints.

**Figure 15 materials-16-04002-f015:**
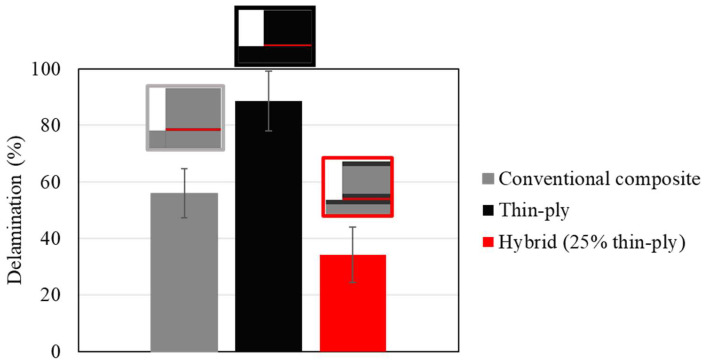
Average delamination ratios for reference conventional composite, thin-ply, and hybrid (25% thin ply) joints.

**Figure 16 materials-16-04002-f016:**
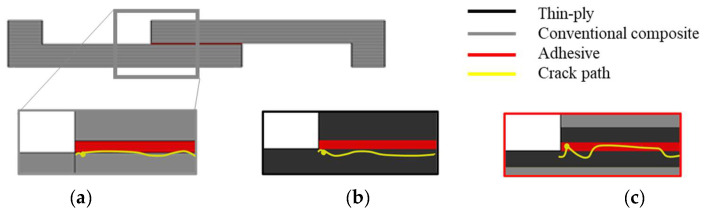
Schematic representation of the failure mechanism for (**a**) reference conventional composite, (**b**) thin-ply, and (**c**) hybrid (25% thin ply) joints.

**Table 1 materials-16-04002-t001:** Main mechanical properties of “AF 163-2k” [41].

Mechanical Property	Value
Young’s modulus [MPa]	1521.87
Shear modulus [MPa]	563.67
Tensile strength [MPa]	46.93
Shear strength [MPa]	46.93
$G_{IC}$ [N/mm]	4.05
$G_{IIC}$ [N/mm]	9.77

**Table 2 materials-16-04002-t002:** Conventional composite mechanical properties [42].

Mechanical Property	Value
$E_{1}$ [MPa]	109,000
$E_{2}$ [MPa]	8819
$G_{12}$ [MPa]	4315
$G_{23}$ [MPa]	3200
$\upsilon_{12}$	0.34
$\upsilon_{23}$	0.38

**Table 3 materials-16-04002-t003:** Cohesive properties of conventional composite [43].

Property	Value
Tensile strength [MPa]	25
Shear strength [MPa]	13.5
$G_{IC}$ [N/mm]	0.33
$G_{IIC}$ [N/mm]	0.79

**Table 4 materials-16-04002-t004:** Thin ply mechanical properties [44].

Mechanical Property	Value
$E_{1}$ [MPa]	101,720
$E_{2}$ [MPa]	5680
$G_{12}$ [MPa]	3030
$G_{23}$ [MPa]	3030
$\upsilon_{12}$	0.38
$\upsilon_{23}$	0.04

**Table 5 materials-16-04002-t005:** Cohesive properties of the thin ply [44].

Property	Value
Tensile strength [MPa]	35
Shear strength [MPa]	32
$G_{IC}$ [N/mm]	0.76
$G_{IIC}$ [N/mm]	0.83

**Table 6 materials-16-04002-t006:** Representative and average delamination area.

Configuration	Representative [mm^2^]	Average [mm^2^]
Conventional composite	190.52	210.00 ± 32.75
Thin-ply	296.82	332.07 ± 39.65
Hybrid (25% thin ply)	110.83	128.20 ± 36.92

## Data Availability

Not applicable.

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
