# Peer review of "Study of Hybrid Composite Joints with Thin-Ply-Reinforced Adherends"

_materials, 2023, doi:10.3390/ma16114002_

Round 1

Reviewer 1 Report

This work experimentally and numerically studied the mechanical performance of composite single lap joints using toughened adherends reinforced by thin-plies. The paper is good written. The suggested revision for the revise:

- Quality fig. 2 should be improved.

-Why is just one type of composite material used? How can it be used in industry?

-In figure 15, please introduce each colure.

-In order to provide a more comprehensive literature review, the authors should cite and discuss the following relevant papers in their revised manuscript:

Static capacity of tubular X-joints reinforced with fiber reinforced polymer subjected to compressive load. Engineering Structures. 2021 Jun 1;236:112041.

Local joint flexibility of tubular T/Y-joints retrofitted with GFRP under in-plane bending moment. Marine Structures, 2021;77, p.102936.

-Where is the validation result of the FE model against the experimental data?

-Please the mesh size on a mesh model

Author Response

REVIEWER #1: The corrections were made by Green color in clean file.

This work experimentally and numerically studied the mechanical performance of composite single lap joints using toughened adherends reinforced by thin-plies. The paper is good written. The suggested revision for the revise:

  1. Quality fig. 2 should be improved.

Answer: Fig 2 was redrawn and replaced by the previous figure.

  1. Why is just one type of composite material used? How can it be used in industry?

Answer: Two different composites with the commercial reference "Texipreg HS 160 T700" and "NTPT-TP415" were used as the conventional and thin-ply respectively.  Therefore, three configurations were considered in this study, two reference single lap joints with conventional composite or thin-ply used as the adherends and a hybrid single lap. The mentioned configurations were shown in Fig 3 from the manuscript. Additional information was added to the abstract.

  1. In figure 15, please introduce each colure.

Answer: Legend was added to Fig 15 (Fig 16 in the revised manuscript).

  1. In order to provide a more comprehensive literature review, the authors should cite and discuss the following relevant papers in their revised manuscript:

-Static capacity of tubular X-joints reinforced with fiber reinforced polymer subjected to compressive load. Engineering Structures. 2021 Jun 1;236:112041.

-Local joint flexibility of tubular T/Y-joints retrofitted with GFRP under in-plane bending moment. Marine Structures, 2021;77, p.102936.

Answer: The mentioned relevant papers were added to the introduction.

  1. Where is the validation result of the FE model against the experimental data?

Answer: The FE model for each configuration was validated by comparing the numerical and experimentally obtained load-displacement curve. This was shown in Fig 11 (in the revised manuscript). Afterward, the numerical damage initiation and final failure mode were also compared to the experimental ones (comparing Fig 5 with fig 12 and comparing Fig 6 with Fig 13 respectively).

  1. Please the mesh size on a mesh model

Answer: A double and single biased mesh distribution was considered in the x direction for the bondline and the adherends respectively. The minimum and maximum size for the mesh distribution was set  to 0.2 and 0.5 mm respectively. However, a uniform mesh distribution with the size of 0.5 mm was considered for the end tabs (in the x direction). Moreover, a uniform mesh distribution through the thickness (y direction) was considered for all models with a mesh size of 0.2 mm. The explanation was added to the manuscript. Moreover, Fig 1 in the response letter (Fig 10 in the manuscript) was added in order to better understand the explanations above.

Reviewer 2 Report

The English expression might be improved by native English speaker

Author Response

REVIEWER #2: The corrections were made by Red color in clean file.

The paper presents a very comprehensive research study involving the performance of hybrid composite laminate reinforced by thin-plies when used as adherends in bonded single lap joints. The authors of the work have carried out a great deal of experimental research and simulations. The content of the work has been presented in a comprehensive manner. The work deserves special recognition, especially in an increase of about 90% in the failure load in the hybrid joint reinforced by thin-ply laminates compared to the standard-ply laminates. Based on my assessment, this work may have the potential to be accepted for publication in Materials. However, a major revision is required before reconsidering for acceptance. There are some remarks as follows:

  1. In section 2.2, the parameters related to CZM in Table 3 and 5 is lack, such as interfacial stiffness. Should authors add it?

Answer: In this research, solid cohesive elements following the triangular traction separation laws were considered to simulate damage evolution (damage initiation and propagation) which could be represented by the material property of the composites (conventional and thin-ply) including the strength, stiffness, and fracture toughness. Additional explanation was added to Section 4.1.

  1. In section 3.2, Figure 5 shows the images of damage initiation in three type joints. However, the locations of damage initiation are lack of clarity. Should the authors provide clear pictures or mark the key information on the graph?

Answer: Since the final failure for all configurations was a combination of delamination and adhesive failure, the damage initiation in the composite joint (whether it occurred firstly in the adhesive layer or within the adherend) needs to be determined. Accordingly, a high-speed camera was used to record the specimens under load. The adherend and adhesive boundaries were roughly defined by correlating the known specimen’s dimensions and the equivalent image pixels. Damage initiation in the reference conventional composite and thin-ply occurs in the composite adherend. In contrast, for the hybrid (25% thin-ply) joint the damage initiation occurs in the adhesive layer. The explanations were mentioned in the manuscript and Fig 2 in the response letter (Fig 5 in the revised manuscript) was modified in order to mark the key information of the image.

  1. In section 3.3, should authors detailed describe the measure method for delamination area? The delamination area marked in Figure 6 is too cloudy to identify. A compared table is suggested to summarize it.

Answer: The open source "IC Measure" software was used to calculate the delamination area for each joint. The delamination ratio is defined as the delamination area divided by the total bonded area, as presented in equation (1). It has to be mentioned that the total bonded area is constant and equal to 375  for all configurations.  Fig 6 was replaced and Table 1 in the response letter (Table 6 in the revised manuscript) was added to clarify the representative delamination area from Fig 6 and the average delamination area. The manuscript was modified accordingly.

Table 1. Representative and average delamination area

Configuration

Representative ()

Average ()

Conventional composite

190.52

210.00±32.75

Thin-ply

296.82

332.07±39.65

Hybrid (25% thin-ply)

110.83

128.20±36.92

  1. In section 3.4, some key failure characteristics suggest to mark on the microscopic images. The reason for this different failure characteristic should provide reasonable explanations or cite correlated works.

Answer: The mentioned failure mechanism was marked in Fig 3 in the response letter (Fig 7 in the manuscript). Moreover, correlated works were cited.

  1. The description about numerical method is incomplete. Authors should add some information on the simulation modelling, such as, the number of elements, the damage initial criterion. In addition, was the effect of mesh density on the results of FEM calculations studied? It is possible that the mesh density would have an impact on the calculation results. Please comment on this briefly in the text of the paper.

Answer: A double and single biased mesh distribution was considered in the x direction (see Fig 4 in the responce letter) for the bondline and the adherends respectively. The minimum and maximum size for the mesh was considered 0.2 and 0.5 mm. However, a uniform mesh with the size of 0.5 mm was considered for the end tabs (in the x direction). Moreover, a uniform mesh distribution through the thickness (y direction) was considered for all model with a mesh size of 0.2 mm. Fig 4 in the response letter illustrates the mesh distribution mentioned above. Accordingly, about 15000 elements were generated for each numerical model. Triangular traction separation laws were applied to the adhesive layers of the model to simulate damage evolution (damage initiation and propagation). The explanations were added to the manuscript. Moreover, Fig 4 in the response letter (Fig 10 in the manuscript) was added.

  1. The initial load of thin-ply laminates is lower than that of hybrid laminates. How to explain it?

Answer: In the case of composite laminates, the laminates were loaded under transverse tensile loading. On the other hand, talking about the single lap joints, a combination of different stress components was transferred to the composite adherends. Moreover, in the case of composite joints, the presence of the adhesive layer and its contribution to load transfer may affect the failure load.

  1. The English expression might be improved by native English speaker

Answer: The manuscript was revised generally in order to improve the English.

Reviewer 3 Report

Here are some comments that need to be addressed before publication.

·       The abstract must be a single paragraph.

·       The first paragraph of the abstract is like an introduction. It must be just a couple of sentences.

·       The abstract is too poor. What are the investigated parameters?

·       One of the methods to increase the adhesive joints' strength is adding nanoparticles which have been investigated by some researchers such as the below. It needs to be summarized in the introduction section, also in terms of numerical modeling, cohesive modeling has been conducted by them which is necessary to be mentioned in the introduction section.

Trapezoidal traction–separation laws in mode II fracture in nano-composite and nano-adhesive joints

Condition monitoring of crack extension in the reinforced adhesive joint by carbon nanotubes

A cohesive model with a multi-stage softening behavior to predict fracture in nano composite joints

·       What is the difference between the current work and the previous paper published by the authors (reference 36)?

·       How does the thin film explained in Fig. 1 enhance the failure load? Explain in more detail. Also, use a better figure to show the structure. Are just the thin films attached to the upper and lower surfaces of conventional composite as shown in Fig.1? It is not clear.

·       What is the standard for Single lap joint manufacturing? Mention this in the manuscript.

·       In Figure 4, the arrow showing the conventional composite joint is pointing to the wrong curve. Fix it.

·       The quality of Figure 6 is not good enough.

·       Use the same figure caption font for all figures. Revise the captions of figures 11 and 12.

The quality of the English language is good.

Author Response

REVIEWER #3: The corrections were made by Blue color in the clean file.

Here are some comments that need to be addressed before publication.

  1. The abstract must be a single paragraph.

Answer: The comment was applied.

  1. The first paragraph of the abstract is like an introduction. It must be just a couple of sentences.

Answer: The abstract was revised accordingly.

  1. The abstract is too poor. What are the investigated parameters?

Answer: The abstract was revised accordingly.

  1. One of the methods to increase the adhesive joints' strength is adding nanoparticles which have been investigated by some researchers such as the below. It needs to be summarized in the introduction section, also in terms of numerical modeling, cohesive modeling has been conducted by them which is necessary to be mentioned in the introduction section.
  • Trapezoidal traction–separation laws in mode II fracture in nano-composite and nano-adhesive joints
  • Condition monitoring of crack extension in the reinforced adhesive joint by carbon nanotubes
  • A cohesive model with a multi-stage softening behavior to predict fracture in nano composite joints

Answer: The mentioned papers were added to the introduction and Section 4.1

  1. What is the difference between the current work and the previous paper published by the authors (reference 36)?

Answer: In the previous paper different configurations of hybrid composite laminates were investigated under pure transverse tensile loading. The effect of thin-ply thickness per total laminate thickness and different stacking sequences were considered. It was found that hybrid laminates with 25% thin-ply per total thickness of the laminate present the highest transverse tensile strength. Accordingly, the presented study illustrates the effect of using hybrid laminates reinforced by thin-ply (mentioned above) as an adhered in a single lap joint. Additional explanation was added to the introduction.

  1. How does the thin film explained in Fig. 1 enhance the failure load? Explain in more detail. Also, use a better figure to show the structure. Are just the thin films attached to the upper and lower surfaces of conventional composite as shown in Fig.1? It is not clear.

Answer: The study from Ramezani et al [1] has shown that replacing layers of conventional composite in a unidirectional laminate with layers of thin-ply can increase the strength under transverse tensile loading. It was mentioned that this is due to the increase in laminate ductility which could postpone the delamination under transverse tensile loading [1]. Moreover, the experimental observation clearly demonstrated that the presence of thin-plies acts as a barrier against crack propagation. It was found that the use of 25% thin-ply per total thickness of the laminate (12.5% on each top) increased the transverse tensile strength considerably. Additional explanation was added to the manuscript and Fig 1.a was changed accordingly.

  1. What is the standard for Single lap joint manufacturing? Mention this in the manuscript.

Answer: No special standard was considered for this study.

  1. In Figure 4, the arrow showing the conventional composite joint is pointing to the wrong curve. Fix it.

Answer: The figure was edited accordingly.

  1. The quality of Figure 6 is not good enough.

Answer: Fig 6 was replaced and Table 6 was added to clarify the representative delamination area from Fig 6 and average delamination area. The manuscript was modified accordingly.

  1. Use the same figure caption font for all figures. Revise the captions of figures 11 and 12.

Answer: The comment was applied precisely (Fig 12 and 13 in the revised manuscript).

Round 2

Reviewer 1 Report

The paper suggests for publication.